# Chemical, Thermal and Mechanical Characterization of Licorice Root, Willow, Holm Oak, and Palm Leaf Waste Incorporated into Maleated Polypropylene (MAPP)

**DOI:** 10.3390/polym14204348

**Published:** 2022-10-15

**Authors:** Serena Gabrielli, Miriam Caviglia, Genny Pastore, Enrico Marcantoni, Francesco Nobili, Luca Bottoni, Andrea Catorci, Irene Bavasso, Fabrizio Sarasini, Jacopo Tirillò, Carlo Santulli

**Affiliations:** 1Chemistry Division, School of Science and Technology, Università degli Studi di Camerino, ChIP Building, Via Madonna delle Carceri, 62032 Camerino, Italy; 2School of Biosciences and Veterinary Medicine, Università degli Studi di Camerino, Via Gentile III da Varano, 62032 Camerino, Italy; 3Department of Chemical Engineering Materials Environment, Sapienza—Università di Roma, Via Eudossiana 18, 00184 Roma, Italy; 4Geology Section, School of Science and Technology, Università degli Studi di Camerino, Via Gentile III da Varano 7, 62032 Camerino, Italy

**Keywords:** lignocellulosic materials, polymer matrix composites, licorice root, palm leaf, holm oak, willow, mechanical properties, thermal properties

## Abstract

The effect of four lignocellulosic waste fillers on the thermal and mechanical properties of biocomposites was investigated. Powdered licorice root, palm leaf, holm oak and willow fillers were melt compounded with polypropylene at two different weight contents, i.e., 10 and 30, and then injection molded. A commercially available maleated coupling agent was used to improve the filler/matrix interfacial adhesion at 5 wt.%. Composites were subjected to chemical (FTIR-ATR), thermal (TGA, DSC, DMA) and mechanical (tensile, bending and Charpy impact) analyses coupled with a morphological investigation by scanning electron microscopy. Although similarities among the different formulations were noted, holm oak fillers provided the best combination of thermal and mechanical performance. In particular, at 30 wt.% content with coupling agent, this composite formulation displayed remarkable increases in tensile strength and modulus, flexural strength and modulus, of 28% and 110%, 58% and 111%, compared to neat PP, respectively. The results imply that all these lignocellulosic waste fillers can be used successfully as raw materials for biocomposites, with properties comparable to those featured by other natural fillers.

## 1. Introduction

Lignocellulosic waste is widely available as the by-product of some industrial sectors, such as furniture manufacturing, papermaking, or food production, but also as the effect of some necessary operations aimed at the effective maintenance of environmental sites, such as woods, forests, parks, etc. In most cases, the biomass obtained after these procedures is neither compostable, nor can produce a significant amount of energy to justify an incineration process, yet it may be considerably hygroscopic [1]. The EU directive 2008/98 suggests not to modify the hierarchy of waste disposal options, stating in practice that, wherever possible, the use of this biomass in a productive system as a secondary raw material needs to be attempted, for example in a wood replacement product, such as particleboards (see e.g., [2]). However, a complete characterization of the material obtained through the introduction of the lignocellulosic waste in a polymer matrix is necessary to prove its suitability for the envisaged use [3]. The primary issue in this regard is the compatibility between the hydrophobic matrix and the hydrophilic filler, and to obtain a sufficient interfacial strength between the two phases.

Maleated polypropylene (MAPP) is a compatibilized thermoplastic, normally added to pure polypropylene (PP), which has been often used in combination with lignocellulosic materials to produce polymer biocomposites [4]. The most common natural fibers have been reported to be successfully introduced in MAPP, such as the case of hemp [5], jute [6], flax [7], coir [8], in the latter case offering also to the fiber a possible penetration into the automotive sector. In addition, MAPP also offered a sufficient compatibility with different types of lignocellulosic waste. In particular, when using MAPP, composites including lignocellulosic agricultural waste, such as sunflower husk, offered a much more effective interfacial adhesion with the matrix, resulting in improved hardness and notched impact strength to the composite [9]. Significant effects were also observed on the modification of thermal degradation profile of the polymer via filling it with by-products, such as oil palm empty fruit bunch (OPEFB), in which case a polypropylene waste matrix was selected [10].

In practical terms, MAPP was found to be adapted to fillers with the most different chemical (cellulose, hemicellulose, lignin, etc.) composition and was able to extend the profile of application of the lignocellulosic waste composites. Of course, this requires a chemical, thermal and mechanical characterization of the fillers “as received” and of the obtained composites.

In this work, a number of lignocellulosic waste types are comparatively considered and characterized into their introduction in MAPP to evaluate their compatibility for use in a semi-structural composite. They have been considered for their availability and their different chemical characteristics to offer a wider perspective on their respective suitability for use in thermoplastic composites. The potential for application of the different waste elucidated so far in the literature has been quite limited. In particular, holm oak has been considered as an alternative source for the production of cellulose pulp [11], willow flour has been proposed as filler for polyethylene matrix [12] and poly(lactic acid) into microcellular foam [13]. On the two remaining fillers, licorice root and palm oil waste, a study concerning their characterization before and after introduction in a limited amount into a poly(urethane-acrylate) (PUA) matrix has been carried out [14]. It is suggested that for all of the fillers considered, working with MAPP would enable their use as the matrix reinforcement in significant amounts, thus providing scope for further application.

## 2. Materials and Methods

### 2.1. Materials

Four different types of lignocellulosic waste were used in this study, namely willow (W), holm oak (HO), palm leaf (P) and licorice root (L). These raw materials were sourced locally considering urban waste and forestry materials available every day, as the most common source of biomass. Licorice root was provided by Amarelli & Fallani, Rossano Calabro, Italy. Biomass from neatening palm leaf from parks and inner-city areas was provided by Ecoflora2 (Ardea, Italy). Willow and holm oak from marginal rural mountain areas were collected under the supervision of the authors.

Polypropylene (PP) was supplied by Repsol (Milan, Italy) under the trade name of ISPLEN PP094, while the coupling agent (CA) was a commercial product named Polybond^®^ 3200 by Addivant Corporation (Danbury, CT, USA), which is a maleic anhydride modified polypropylene homopolymer with a high maleic anhydride content in the range 0.8–1.2%.

### 2.2. Biomass Chemical Composition Analysis and Characterization

For the qualitative determination and chemical characterization of biomass components, an initial Fourier Transform spectroscopic analysis in the MID-IR region was performed, being a rapid and non-destructive technique. FTIR-ATR analysis was performed with a Perkin-Elmer (Beaconsfield, UK) FTIR spectrometer Spectrum Two UATR, equipped with ZnSe crystal. The measurements were performed in a 400–4000 cm^−1^ range at a 2 cm^−1^ resolution, 4 scans and processed by a Perkin-Elmer data manager. Measurements were performed three times per sample.

Lignocellulosic biomasses were characterized as total content and in Figure 1, the overlap of the four different FTIR spectrums was reported. They are mostly composed of lignin, hemicellulose, and cellulose, but there are small differences between the peak proportions due to their structural complexity.

The major absorption peaks are chosen and shown (Figure 1): the -OH stretching signified by very strong peaks at 3200–3500 cm^−1^, stretching of aliphatic –CH and –CH_2_ and –CH_3_, between 2800–3000 cm^−1^. The broad peak, very small in intensity, at around 1730 cm^−1^, shows a very low holocellulose to lignin ratio. The absorption peak at around 1620 cm^−1^ is characteristic of aromatic skeletal vibration associated to the out of plane stretching of aromatic groups, the absorption peaks in the region from 950 to 1200 cm^−1^ are assigned to the –C–O stretching vibration in holocellulose, and the peak at 1420 cm^−1^ is assigned to the –CH, –CH_2_ bending of aliphatic carbons. The quite strong peak at 1320 cm^−1^ is assigned to stretching and deformation vibrations of the –C–H group in the glucose unit of cellulose structure [15].

Then, the chemical composition of the raw lignocellulosic flours was determined, and expressed as the percentage content of lignin, α-cellulose, and hemicellulose, (see Table 1). A minimum of three times testing were performed on each sample, and their average value was calculated. This work was performed with the aim of finding a correlation, based on the chemical composition of the different samples, with the mechanical properties of the resulting composites. On the other side, it can be noticed that the markedly different behavior, when limiting observations to the chemical composition, would only be expected from palm leaf filler, which has a much lower amount of cellulose, whereas the three other sources of biomass express comparable chemical composition. In contrast, the amount of hemicellulose is almost the same for all biomasses. From overlapping, it is noticeable that the IR spectrums of all samples provide significant evidence, including changes in chemical arrangement, and functionalization.

Lignin extraction was carried out directly from biomass flours through a dissolving method [16,17]. The remaining solid residue was washed with acetone to obtain separated cellulose [18,19]. The last fraction, hemicellulose, was isolated by alkaline extraction [20,21]. The detailed methodology employed for the extraction of the different components is reported in the Supplementary information provided in Appendix A.

### 2.3. Composite Processing

Composite materials were prepared using a two-step process, consisting of dry mixing followed by melt compounding. Woody fillers were previously ground and reduced to powder with a ball vibro-mill, Retzsch MM 400 (Retsch GmbH, Haan, Germany) for size reduction and homogenization, with a 35 mL stainless steel chamber equipped with stainless steel grinding balls with diameter of 20 mm. The grinding frequency for the grinding process was 20 Hz for 15 min. Figure 2 shows the appearance of the fillers after grinding.

Prior to extrusion, materials were dried overnight at 80 °C. A co-rotating twin screw extruder (Thermo Scientific Process 11, Thermo Fisher Scientific, Waltham, MA, USA) was used for melt compounding of the dry mixed blends. A decreasing temperature profile with a maximum barrel temperature of 215 °C and a die temperature of 210 °C was used, while the screw speed was kept at 150 rpm. The specimens for mechanical characterizations were obtained by injection molding (Haake MiniJet II Pro, Thermo Fisher Scientific). The molten material was transferred by a heated cylinder in the injection molding machine. The mold was kept at 60 °C, while the loading cylinder was heated at 200 °C. The injection procedure consisted of two steps: a first injection step at a pressure of 650 bar for 10 s, and a post injection step at 200 bar for additional 10 s.

In this work, specimens have been manufactured according to ISO 178, ISO 179-2 and ISO 527-2 for bending, Charpy impact, and tensile tests, respectively.

All of the composite formulations are listed in Table 2.

### 2.4. Characterization of Injection Molded Composites

#### 2.4.1. Mechanical Testing

Tensile tests were performed according to ISO 527-2 on a Z010 universal testing machine with a 10 kN load cell (Zwick/Roell, Ulm, Germany) in displacement control. Tests were carried out at a rate of 10 mm/min with type 1 BA samples until fracture, and the strain was measured with a contacting extensometer (gauge length = 30 mm). Five replicates were performed on each formulation.

The same equipment was used for the three-point bending tests according to ISO 178, at a rate of 5 mm/min and with a support span length of 64 mm. Strain was accurately measured with a displacement transducer in contact with the samples. Five replicates were performed on each formulation.

ISO 179-2 was used as standard for notched (type A) Charpy impact tests in an edgewise mode. A span of 62 mm was used, and tests were conducted on a CEAST/Instron 9340 (Pianezza, Italy) instrumented drop weight tower by using an impact velocity of 2.90 m/s. At least five replicates were performed on each formulation.

#### 2.4.2. Thermal Testing

A Setsys Evolution system by Setaram (Caluire, France) was used to determine the thermal stability of composites by thermogravimetric analysis (TGA). Samples of around 40 mg were placed in an alumina pan and were analyzed in a nitrogen atmosphere from 25 °C to 800 °C with a heating rate of 10 °C/min.

A differential scanning calorimeter DSC 214 Polyma by Netzsch GmbH (Selb, Germany) under a constant nitrogen flow of 50 mL/min was used to investigate the crystallization and melting behavior of the different formulations. Samples of 9.0 to 10.0 mg were placed in a concavus aluminum crucible with pierced lid and were analyzed according to the following thermal program: heating from −40 °C to 220 °C (5 min hold), cooling to −40 °C (10 min hold), and heating to 220 °C, all steps conducted with a rate of 10 °C/min. The degree of crystallinity (*X_c_*) of the samples was calculated according to Equation (1):(1)$$X_{c}\;\left(\%\right)=\frac{\Delta H_{m}}{\Delta H_{m0}}\times \frac{100}{\left(1-w_{f}\right)},$$
where Δ*H_m_* is the experimental enthalpy of melting of the sample (J/g), Δ*H*_*m*0_ the enthalpy of melting for 100% crystalline PP (J/g) taken as 209 J/g [22], while *w_f_* is the weight fraction of filler in the composite formulation.

Dynamic mechanical analysis (DMA) was performed in a three-point bending mode on a DMA 242 E Artemis by Netzsch GmbH. Samples (60 × 10 × 4 mm) were subjected to a heating rate of 2 °C/min from −100 °C up to 130 °C at a frequency of 1 Hz.

For all thermal characterizations, three replicates were performed on each formulation.

### 2.5. Morphological Characterization

The fracture surfaces of samples after tensile and Charpy impact tests were imaged by a field-emission scanning electron microscope (FE-SEM) Mira3 by Tescan (Brno, Czech Republic). Samples were sputter-coated with gold prior to analysis.

## 3. Results

### 3.1. Filler Morphology and Thermal Stability

SEM analysis was used to image the morphology of the different natural fillers. Figure 3 shows the SEM micrographs of the fillers at different magnifications.

As can be seen, all lignocellulosic fillers after the milling process featured irregular morphologies, consisting of long fibers, thinner fibers/fibrils, and non-fibrous materials in the form of particulates. From SEM micrographs, the length and diameter of around 70 fillers for each waste material were measured to evaluate the average aspect ratio (length/diameter). In all cases, it was found to be lower than 5, with licorice root displaying a more refined structure compared to the other waste materials. The four different fillers can be ranked in the following decreasing order concerning the average aspect ratio (in parenthesis): licorice root (3.3) > palm leaf (2.9) > willow (2.7) > holm oak (2.3). The reduced aspect ratio is expected to hinder the reinforcing efficiency of the fillers according to Halpin-Tsai model and to the lower filler/matrix interfacial area available. In addition, the occurrence of large particulates might introduce defect points instead of granting a better distribution of the applied load as happens for high aspect ratio fibers [23].

Figure 4 shows typical thermogravimetric (TGA) and derivative thermogravimetric (DTG) curves of the powdered lignocellulosic fillers.

As widely discussed in the literature, the initial weight loss is attributed to the removal of moisture content from the fillers [24], though it was higher in palm leaf compared to the other fillers. The lignocellulosic fillers were not pre-dried before performing the TGA analysis, and this explains the occurrence of this first mass loss at temperatures lower than 150 °C, where free water evaporates at a lower temperature while the bound water from chemical bonds with the hydroxyl groups present in hemicellulose and cellulose evaporates at higher temperatures [25]. The onset of thermal degradation, defined as the temperature where 5 wt.% loss occurs, was significantly different from palm leaf compared to the other types of lignocellulosic waste, as reported in Table 3.

The lower thermal stability of palm leaf was confirmed also with increasing temperature, with a maximum degradation temperature at 335 °C, around 20 °C lower than the other fillers. All of these values are in line with those exhibited by other common natural fibers [26]. After 250 °C, a sudden weight loss was observed, indicating the decomposition of hemicellulose and glycosidic linkages of cellulose. When comparing the fillers, it is worth noting that palm leaf filler featured two well-separated degradation peaks in the range 250–350 °C [27], a first peak around 290 °C and a second peak at 355 °C, which can be assigned to the thermal decomposition of hemicellulose and cellulose, respectively [28].

This behavior is different from that usually observed with other natural fillers [29,30,31,32,33], which is more similar to that featured by holm oak, willow fillers, and to a lower degree, by licorice root waste. The different intensity of the main cellulose peak degradation suggests a lower cellulose content in palm leaf compared to the other fillers, as confirmed by chemical analysis (Table 1). The high-temperature tail shown in Figure 4 is due to the degradation of lignin [34], which resulted in a higher charred organic material residue for palm leaf with respect to the other lignocellulosic fillers.

### 3.2. Thermal Properties of Injection Molded Composites

Thermogravimetric analysis provided information about the thermal stability of PP as a function of an increasing number of natural fillers. The results of TGA and derivative of the thermogravimetric analysis (dTG) for all composite formulations are shown in Figure 5.

Average parameters for thermal degradation, such as onset temperature at 5% weight loss (T_d5_), temperature at 10% weight loss (T_d10_), and maximum degradation temperature (T_max_) are summarized in Table 4.

The thermal degradation of the neat PP starts at approximately 427 °C and reaches its maximum at around 465 °C. As can be inferred from Figure 5 and Table 4, all of the composite formulations feature an intermediate behavior between PP and natural fillers. Regardless of filler type, the introduction of lignocellulosic fillers in neat polymer resulted in a reduction in its thermal stability, a trend that increased with increasing amount of filler. The PP weight loss occurred in a one-step degradation process from 420 to 500 °C, by random scission and thermal depolymerization of weak sites of the PP main chains [35]. It can be noted that the decomposition of composites is much more complicated, with a lower onset temperature of decomposition in the range 200 °C to 360 °C due to the degradation of hemicelluloses and cellulose [36]. The most thermally stable composite material was the one incorporating holm oak waste, while the lowest thermal stability, particularly evident at the maximum weight content, was displayed by composites featuring palm waste as filler, thus confirming results reported in Table 3. The presence of coupling agent was found not to affect the overall thermal degradation profile, but considering a constant amount of filler, it resulted in a reduced thermal stability compared to untreated formulations, except for licorice root filler. This result is in contrast with other studies, where usually the use of a maleated compatibilizing agent improved the thermal stability of the composites, due to the improved interfacial adhesion and additional intermolecular bonding through esterification reaction between hydroxyl groups of lignocellulosic fillers and anhydride functional group of maleated PP [36]. However, it is not totally unexpected to see a decrease in thermal stability, as it has been demonstrated elsewhere that the amount of compatibilizing agent in a composite needs to be optimized depending on the type of filler, and when used beyond this threshold, it might induce a dispersion effect rather than a coupling effect at the filler/matrix interface [37,38]. This leaves room for further improvements. Another important result is that the onset temperature of thermal degradation is well above 200 °C for all composites, confirming the feasibility of their production by extrusion and injection molding.

The thermal transitions of neat PP and its composites were evaluated by DSC, analyzing the melting temperature (T_m_), the crystallization temperature (T_c_), and the enthalpy of fusion, which was used for determining the crystallinity index (*X_c_*) (Table 5). The thermograms (not shown) obtained for the neat matrix and for all composites were conventional and very similar to each other.

The values of melting temperature and associated enthalpy were obtained from the second heating scan while the crystallization temperature was calculated from the cooling curve. The dynamic crystallization behavior did not highlight any positive effect of the fillers on the crystallization kinetic of polypropylene. On the contrary, irrespective of filler type, a decrease in the crystallization temperature was detected but with a very limited modification of the perfection of matrix crystallites, which is usually ascribed to the heterogeneous nucleation promoted by the presence of the lignocellulosic fillers, as confirmed by the small variations in terms of melting temperature [39]. It is interesting to note that, with the exception of the licorice root fibers, the crystallinity fraction of the remaining composite formulations increased compared to the neat matrix, in contrast with other lignocellulosic agricultural wastes [40,41,42]. In this regard, the best performance is offered by holm oak fillers. In addition, the coupling agent played a role and provided the fillers surface with chemical features responsible for nucleating the transcrystallinity [36], without decreasing the regularity of the matrix molecular chains and corresponding packing efficiency. It has been shown that maleated PP is mainly located at the interfacial region [43] and the strong interaction between the lignocellulosic fillers and the matrix can provide transcrystallinity effect [44]. With increasing filler content and presence of coupling agent, a decrease in the degree of crystallinity was measured, though it remained higher than the neat PP. This effect might be due to the presence of excessive density of nuclei induced by the higher amount of filler/matrix interfacial area available, which restricts the crystal growth around the fillers with the growing front impinging quickly with spherulites nucleated in the bulk.

### 3.3. Dynamic Mechanical Behavior of Injection Molded Composites

DMA was used to assess the effect of filler species on the viscoelastic properties and fiber/matrix interaction of composite formulations as a function of temperature. The results are displayed in Figure 6 in the form of thermograms of the storage modulus (E’) and tanδ. Table 6 lists the glass transition temperature (T_g_) evaluated as tanδ peak, and the E’ values at different temperatures.

With increasing temperature, the storage modulus values of neat PP and composite formulations decreased as the matrix softened, but the reduction in matrix modulus in composites was partially compensated by the stiffness of the fillers, especially at filler amount of 30 wt.%. All of the fillers provided higher values of E’ compared to untreated PP, with the best performance offered by licorice root fillers, suggesting that the degree of crystallinity is less important than the filler/matrix interfacial adhesion in governing the evolution of storage modulus with temperature [45]. This is supported by the highest level attained by the storage modulus with the addition of coupling agent, in all composite formulations. It is believed that the maleated polypropylene causes a strong interaction between the fillers and PP matrix, likely creating a stronger and stiffer interfacial layer.

The tan δ curve of neat PP is usually characterized by three relaxations: the α transition around 100 °C, the β transition around 10 °C, and the γ transition around −80 °C [46]. γ transition, characterized by localized bond movements (bending and stretching) and side chain movements, is not clearly defined in the thermograms of Figure 6, because it is very close to the starting temperature of the DMA test. Indeed, two transitions were clearly noted, one around 0 °C and the other around 80 °C. The peak close to 0 °C (β relaxation) corresponds to the glass transition temperature of PP, where relaxation of unrestricted amorphous PP chains occurs. The first characteristic for all of the investigated materials is the lack of a significant change in the glass transition temperature. In fact, the main relaxation peak occurs for all specimens in a quite narrow temperature range of 3–6 °C. This small variation might be due to the plasticizing effect of moisture in composite samples containing hydrophilic lignocellulosic fillers.

However, with increasing filler content, tan δ peak values dropped in filler/compatibilized PP composites. The height and area under the tanδ curve indicate the total amount of energy that can be dissipated by a material, therefore a large area points toward a high degree of molecular mobility and higher damping properties. It is believed that for composite samples, the reduction in peak area, particularly evident for formulations reinforced with holm oak and palm leaf fillers, suggests a restricted mobility of PP molecules in the relaxation process, due to the stronger filler/matrix interfacial adhesion [40,47,48]. In this regard, the modification of PP matrix with the maleated coupling agent resulted in being particularly effective. The α transition is linked to the relaxation of the rigid amorphous PP chains in the crystalline phase [49], which can occur by a lamellar slip mechanism and rotation in the crystalline phase [50]. In all composite samples, the α relaxation region displayed a gradual decrease in the peak amplitude with increased amount of fillers and coupling agent, suggesting that the lamellar movement in the crystalline phase is strongly affected and hindered by the presence of the fillers, as observed in other studies [35,40,47].

### 3.4. Mechanical Properties of Injection Molded Composites

Tensile strength and modulus values of the composites (Figure 7) were within the ranges of 19.9–32.8 MPa and 1.1–2.3 GPa, respectively, and compared favorably with those found in the literature for other natural fillers, as summarized in Table 7.

The composite formulations displayed a macroscopic ductile behavior at 10 wt.% of fillers, but this tendency was reduced by increasing filler content above 10 wt.% and by adding the coupling agent, as can be seen in Figure 8. In all cases, for both non-compatibilized and compatibilized systems, the Young’s modulus was enhanced compared to neat PP, which is a quite common result in polymer matrix composites, as the stiff lignocellulosic fillers are able to hinder the molecular mobility of polymer chains [47]. This confirms the results from DMA tests. Apart from holm oak fillers, in non-compatibilized formulations, a decrease in tensile strength with increasing filler content was noted, which is ascribed to a lack of interfacial adhesion with the hydrophobic PP, only partially balanced by an increase in crystallinity (Table 5). Indeed, licorice root fillers, characterized by the lowest crystallinity fraction compared to the neat PP, featured the lowest tensile strength among all configurations. This behavior has been observed in many other studies dealing with fossil-based [42] and renewable polymer matrices [51,52,53]. Holm oak fillers featured a different trend, with an increase in tensile strength over the baseline even in non-compatibilized composites, due to the higher crystallinity degree in the resulting formulations (Table 5). The coupling agent proved to be effective for all four fillers investigated in the present study, with the best performance offered by composites reinforced with holm oak and willow fillers.

**Table 7 polymers-14-04348-t007:** Summary of mechanical properties of polymer matrix composites reinforced with natural fillers.

Matrix Type	Filler Type	Coupling Agent	Tensile Modulus (GPa)	Tensile Strength (MPa)	Flexural Strength (MPa)	Flexural Modulus (GPa)	Filler Content (wt%)	Reference
PP	Yerba mate	no	0.8	22.5	39.4	2.2	30	[54]
PP (recycled)	Yerba mate	yes	0.6	23.7	35.4	1.8	30	[55]
PP	Buckwheat husk	no	1.6	18.5	-	-	30	[47]
PP	Wood flour	no	2.5	26.0	-	-	30	[47]
PP	Buckwheat husk	yes	1.6	29.0	-	-	30	[47]
PP	Wood flour	yes	2.2	35.5	-	-	30	[47]
PP	Flax	yes	1.1	27.5	49.5	1.7	30	[56]
PP	Poplar	yes	1.4	23.0	37.5	2.4	30	[57]
PP	Rice husk	yes	1.6	26.5	38.0	2.3	30	[57]
PP	Wheat straw	yes	1.5	27.0	42.0	2.5	30	[57]
PP	Corn stalk	no	3.7	26.1	-	-	40	[58]
PP	Corn stalk	yes	3.8	38.5	-	-	40	[58]
PP	Microcrystalline cellulose	no	2.4	32.0	37.5	1.6	20	[59]
PP	Microcrystalline cellulose	yes	2.7	42.0	42.5	1.6	20	[59]
PP	Poplar	yes	5.4	28.2	47.1	5.3	50	[60]
PP	Hemp	yes	2.7	32.0	49.5	1.8	30	[61]

The covalent bonding between the anhydride group and the hydroxyl groups of the lignocellulosic fillers along with chain entanglement between maleated coupling agent and PP chains are responsible for a sound filler/matrix interface and stress transfer [37,62]. The stronger interfacial adhesion is clearly visible in the SEM micrographs of the fracture surfaces of composites after tensile tests, collected in Figure 9, Figure 10, Figure 11, Figure 12, Figure 13, Figure 14, Figure 15 and Figure 16 for the different fillers. As can be seen in Figure 9, Figure 11, Figure 13 and Figure 15, a ductile failure at the microscopic scale governs the fracture surface of non-compatibilized systems, but it shows more brittle features after the addition of coupling agent which hinders the chains mobility. At the same time, a poor filler/matrix interfacial adhesion in all systems has been detected, characterized by filler pull-out and clear debonding at the filler/matrix interface, with fillers barely wetted by the matrix. In compatibilized systems (Figure 10, Figure 12, Figure 14 and Figure 16), the enhanced interfacial adhesion is well-supported by fillers embedded and wetted by the PP matrix with negligible evidence of gaps at the filler/matrix interface.

The flexural properties displayed similar trends to those of the tensile ones (Figure 17), with stiffness in the range of 1.1–2.1 GPa and strength in the range of 29.6–51.3 MPa, well comparable with other natural fiber composites (Table 7).

Figure 18 shows the impact resistance as a function of filler type and coupling agent. It can be clearly seen that the neat PP matrix has an impact absorption energy much higher than the composites.

Generally, the inclusion of stiff fillers in a ductile matrix results in a significant reduction in fracture toughness [54,63], but the reasons are different and complex in nature, and no conclusive study is available to explain the poor impact performance of composites reinforced with natural fillers [64]. Some possible causes involve the reduced matrix ductility induced by the plastic constraint exerted by the stiff fillers, the stress concentration at filler ends and the poor filler/matrix interfacial adhesion. It can be mentioned that in many situations, the presence of fillers provides positive or negative influence depending on the interfacial adhesion [65]. In the present study, no remarkable differences among composite configurations were observed, but a general trend involving an increase in impact strength with increasing filler content and the presence of coupling agent can be highlighted [55,59,66,67]. The fracture surface showed a brittle character (Figure 9, Figure 10, Figure 11, Figure 12, Figure 13, Figure 14, Figure 15 and Figure 16), but the better filler/matrix adhesion enhanced the work of fracture of the interphase that resisted crack propagation, thus supporting the effectiveness of maleated PP.

## 4. Conclusions

Composites of PP and four ground lignocellulosic waste fillers were successfully produced by extrusion up to 30 wt.% filler content with compatibilization performed by a commercial maleated coupling agent. All the fillers, i.e., licorice root, holm oak, palm leaf and willow, shared a common and irregular morphology with a limited fiber aspect ratio, lower than 5, and a reduced notched Charpy impact strength, when compared to neat PP. On the other side, thermal analysis by TGA highlighted the best thermal stability of composites reinforced with holm oak fillers, while from DSC, a nucleating effect was exhibited by all fillers with the exception of licorice root. The higher crystallinity resulted in an increase in tensile strength over the neat PP even in non-compatibilized holm oak-based composites, in contrast to the remaining formulations. DMA results highlighted a reduction in β-relaxation peak area with increasing filler content and the presence of coupling agent, suggesting a restricted mobility of PP molecules due to the stronger filler/matrix interfacial adhesion in holm oak fillers-based composites. This resulted in materials with the best combination of tensile and flexural properties, with enhancements over the neat PP of 28% and 110% in tensile strength and modulus, respectively, and of 58% and 111% in flexural strength and modulus, respectively. Even willow-based composites featured intriguing mechanical properties, with improvements of 27% and 100% in tensile strength and modulus, respectively, and of 35% and 88% in flexural strength and modulus, respectively. Although some differences were reported, all waste fillers investigated in the present study provided composites with mechanical performance comparable to those of other natural fillers, thus confirming their suitability as a valuable untapped source of raw materials for the production of sustainable and higher value biocomposites.

## Figures and Tables

**Figure 1 polymers-14-04348-f001:**
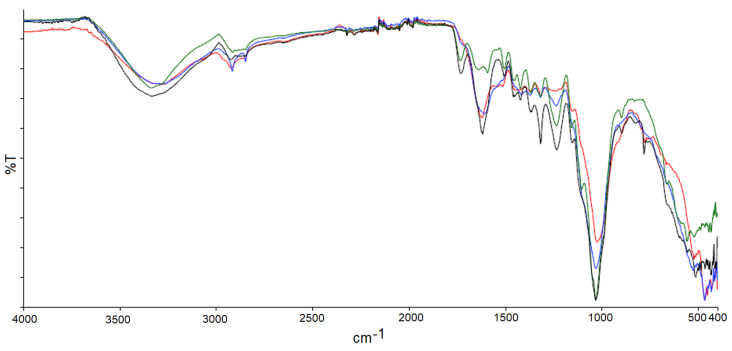
FTIR-ATR overlap for Licorice Root (L, red), Palm Leaf (P, blue), Willow (W, green) and Holm Oak (OH, black).

**Figure 2 polymers-14-04348-f002:**
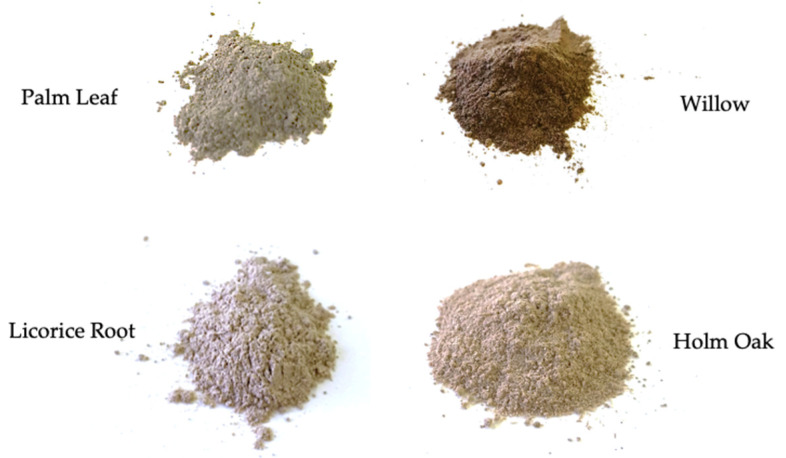
Appearance of ground lignocellulosic fillers.

**Figure 3 polymers-14-04348-f003:**
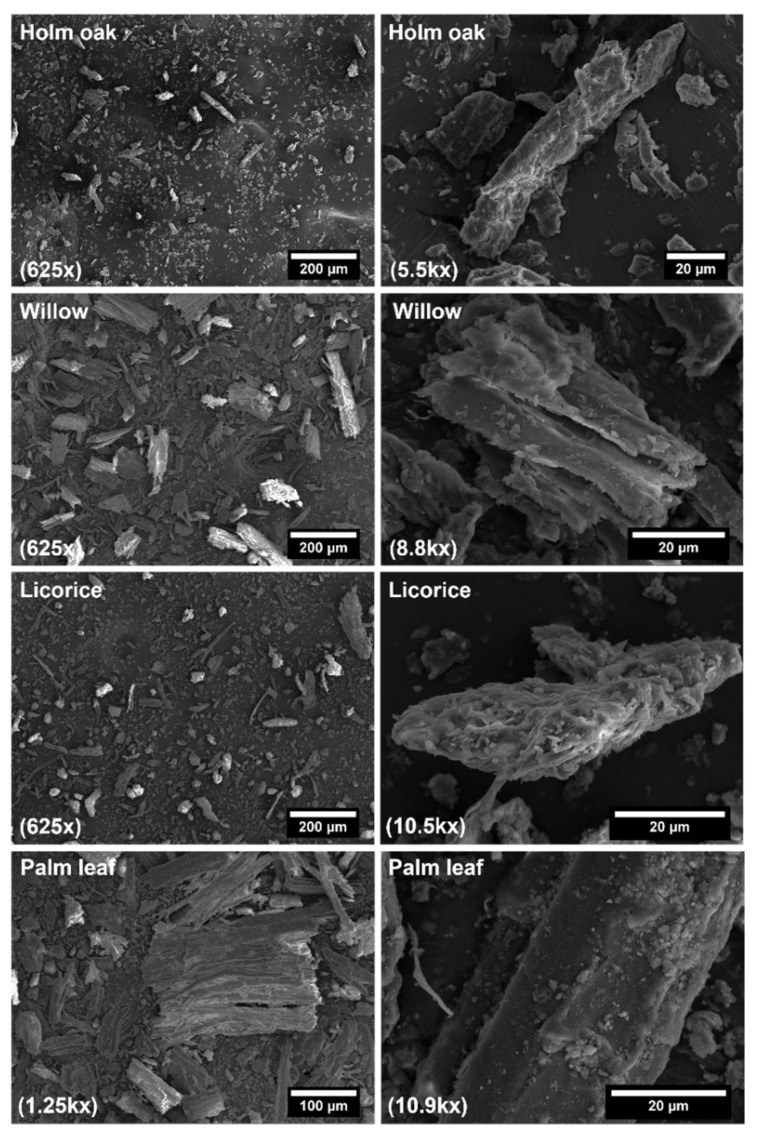
SEM micrographs of ground fillers (in parenthesis the corresponding magnification).

**Figure 4 polymers-14-04348-f004:**
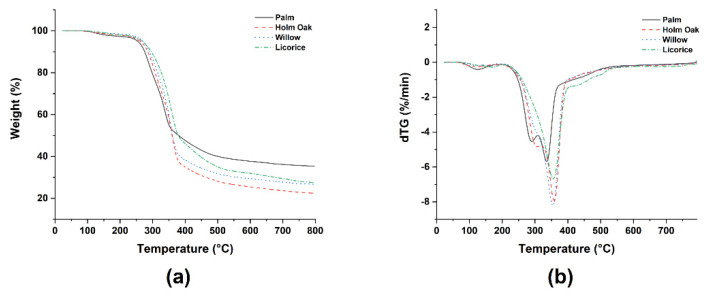
Thermal degradation behavior of lignocellulosic wastes: (**a**) weight loss vs. temperature and (**b**) derivative of weight loss vs. temperature.

**Figure 5 polymers-14-04348-f005:**
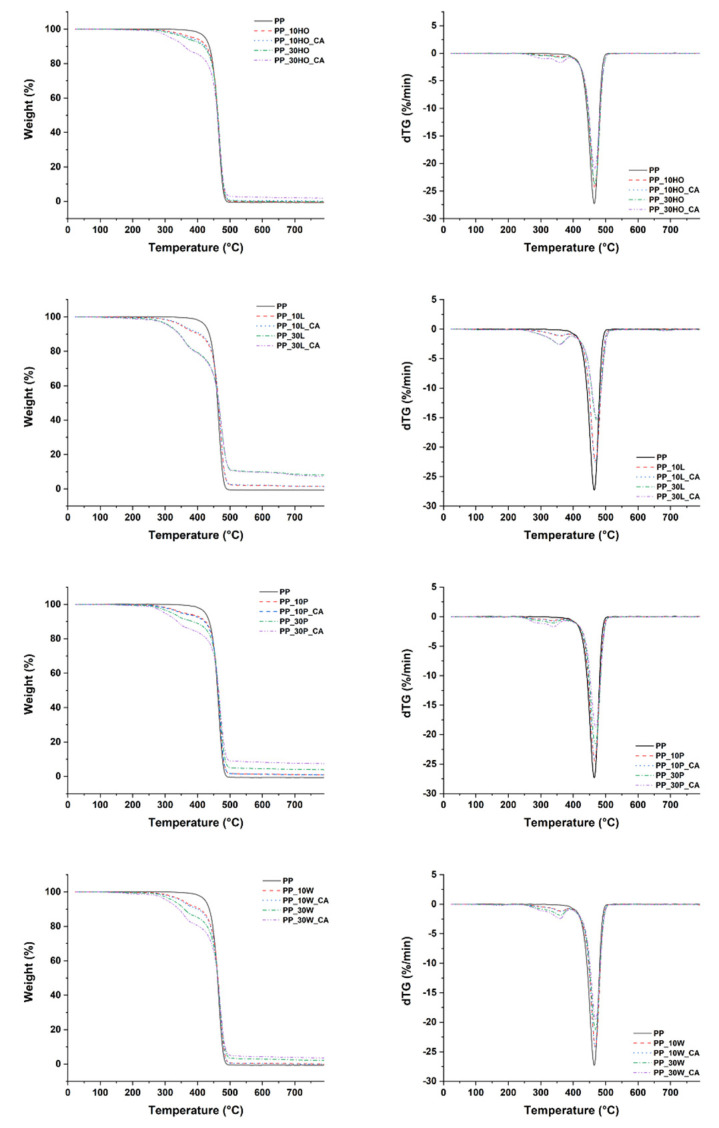
TGA (**left**) and dTG (**right**) curves of PP-based composites.

**Figure 6 polymers-14-04348-f006:**
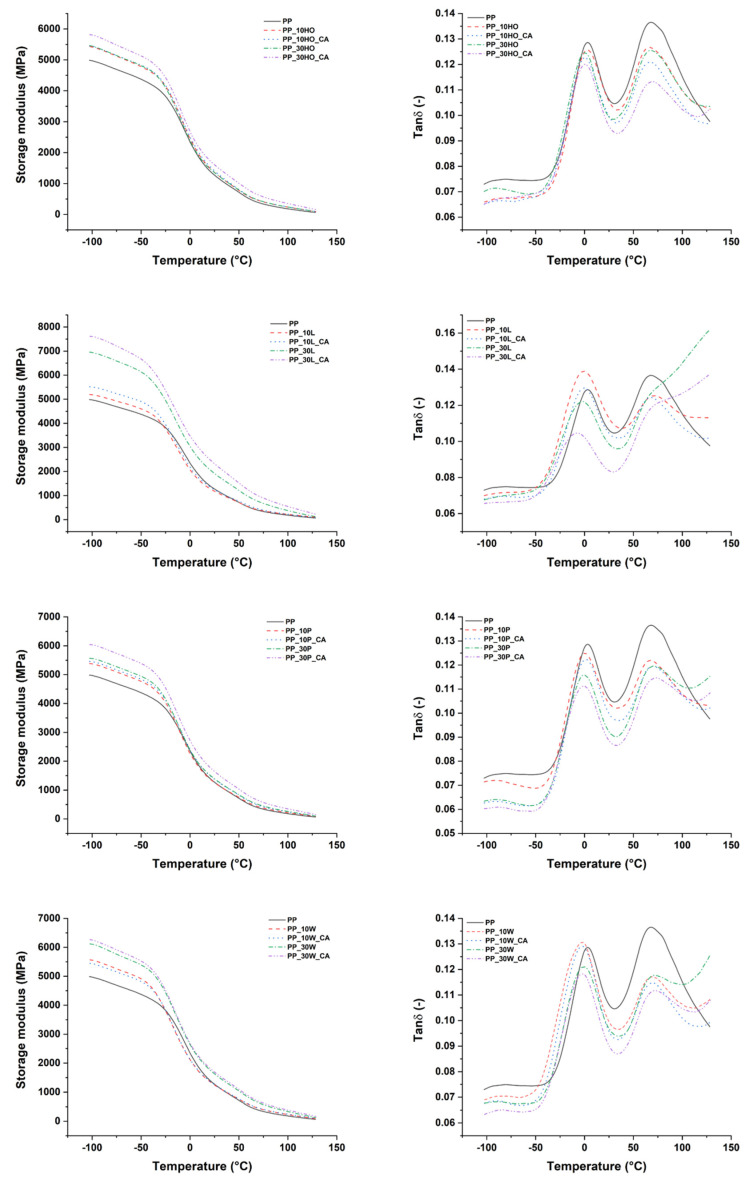
DMA thermograms of storage modulus (**left**) and tan δ (**right**) of PP-based composites.

**Figure 7 polymers-14-04348-f007:**
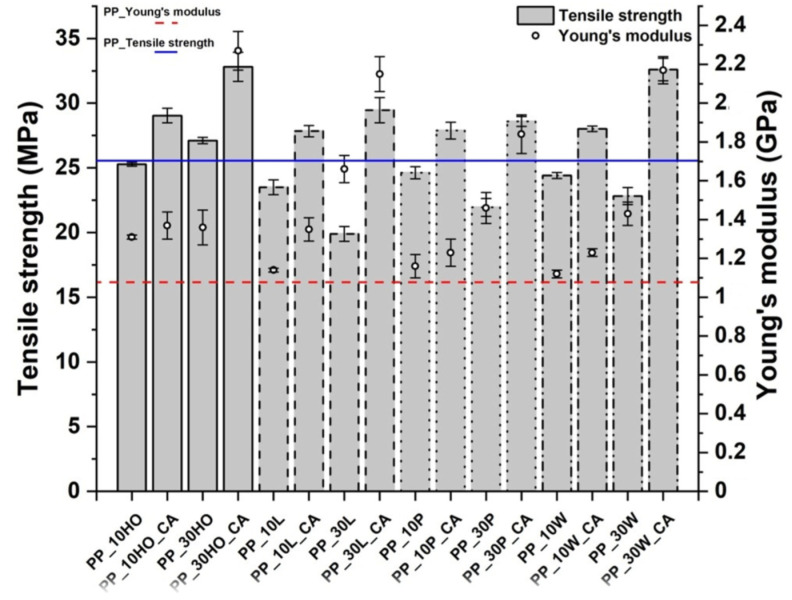
Tensile strength and modulus values of PP-based composites.

**Figure 8 polymers-14-04348-f008:**
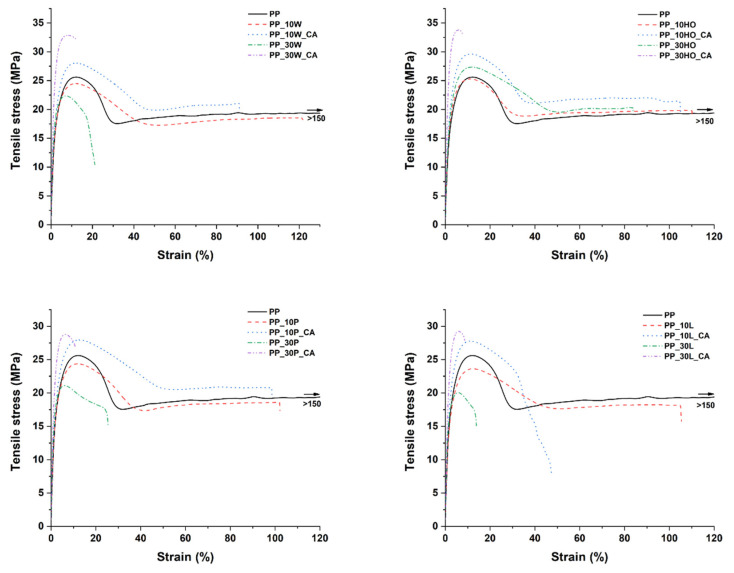
Typical stress–strain curves from tensile tests of PP-based composites.

**Figure 9 polymers-14-04348-f009:**
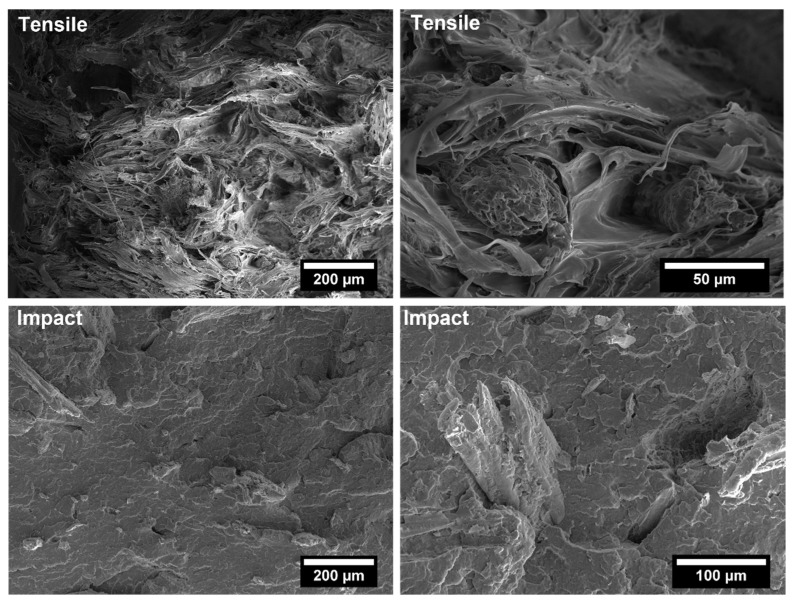
SEM micrographs of PP-based biocomposites with 30 wt.% of holm oak fillers without coupling agent after tensile and Charpy impact tests.

**Figure 10 polymers-14-04348-f010:**
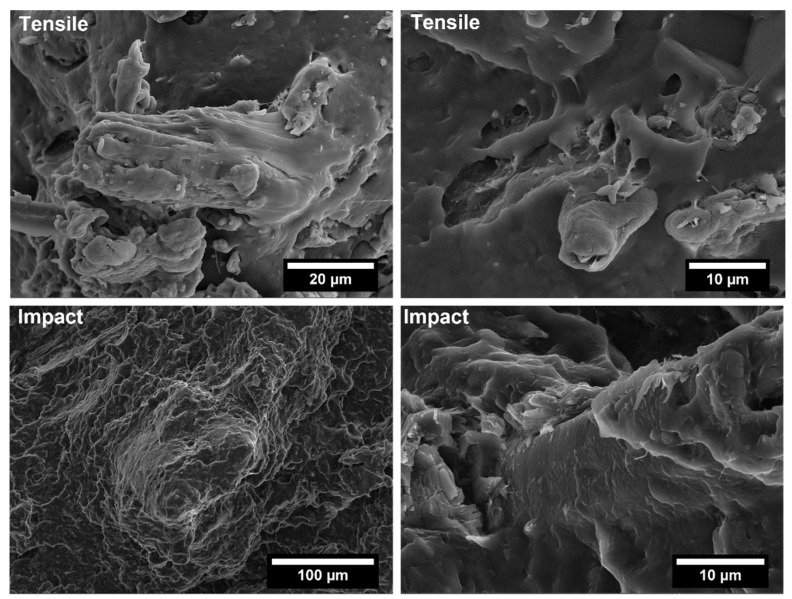
SEM micrographs of PP-based biocomposites with 30 wt.% of holm oak fillers with coupling agent after tensile and Charpy impact tests.

**Figure 11 polymers-14-04348-f011:**
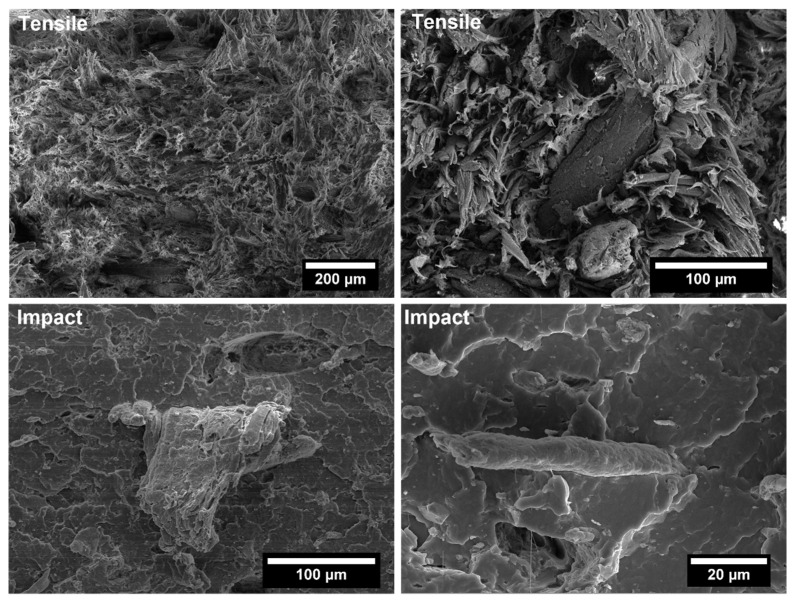
SEM micrographs of PP-based biocomposites with 30 wt.% of licorice root fillers without coupling agent after tensile and Charpy impact tests.

**Figure 12 polymers-14-04348-f012:**
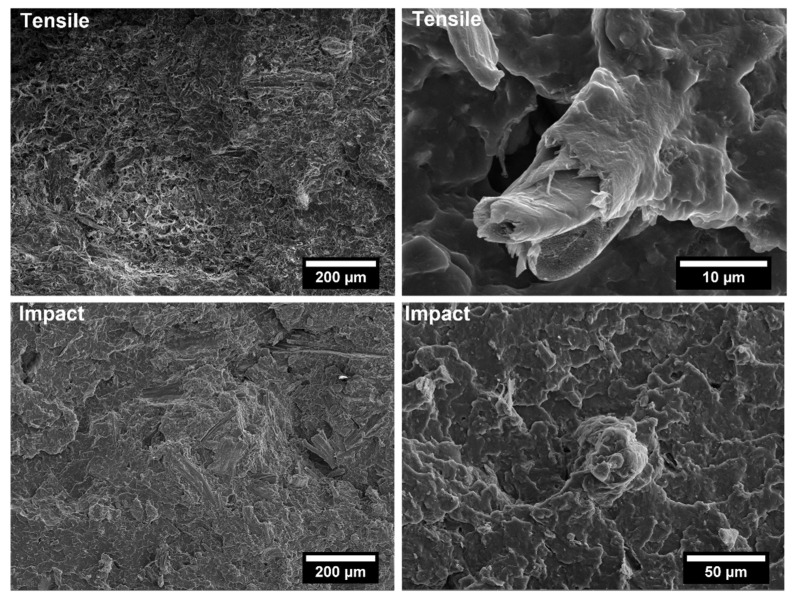
SEM micrographs of PP-based biocomposites with 30 wt.% of licorice root fillers with coupling agent after tensile and Charpy impact tests.

**Figure 13 polymers-14-04348-f013:**
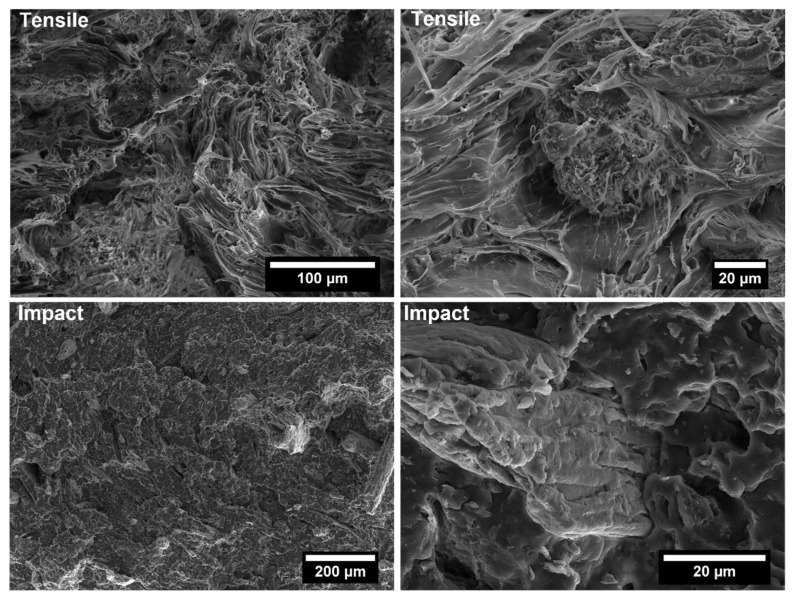
SEM micrographs of PP-based biocomposites with 30 wt.% of palm leaf fillers without coupling agent after tensile and Charpy impact tests.

**Figure 14 polymers-14-04348-f014:**
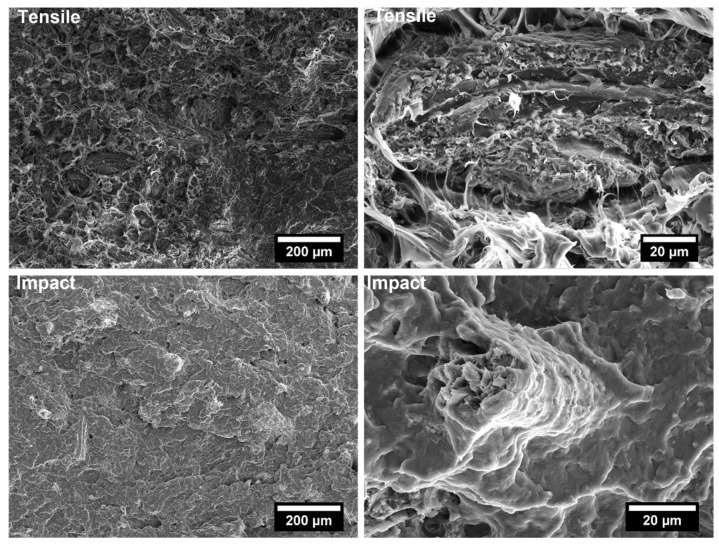
SEM micrographs of PP-based biocomposites with 30 wt.% of palm leaf fillers with coupling agent after tensile and Charpy impact tests.

**Figure 15 polymers-14-04348-f015:**
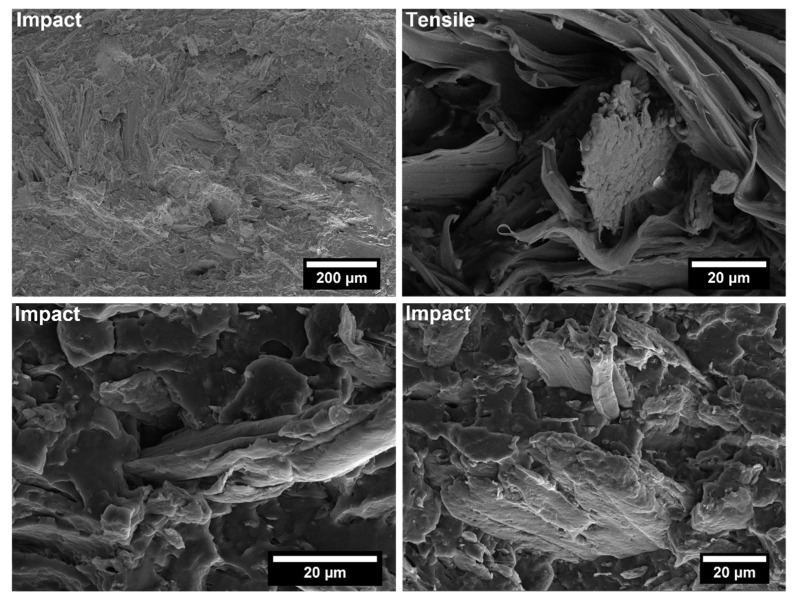
SEM micrographs of PP-based biocomposites with 30 wt.% of willow fillers without coupling agent after tensile and Charpy impact tests.

**Figure 16 polymers-14-04348-f016:**
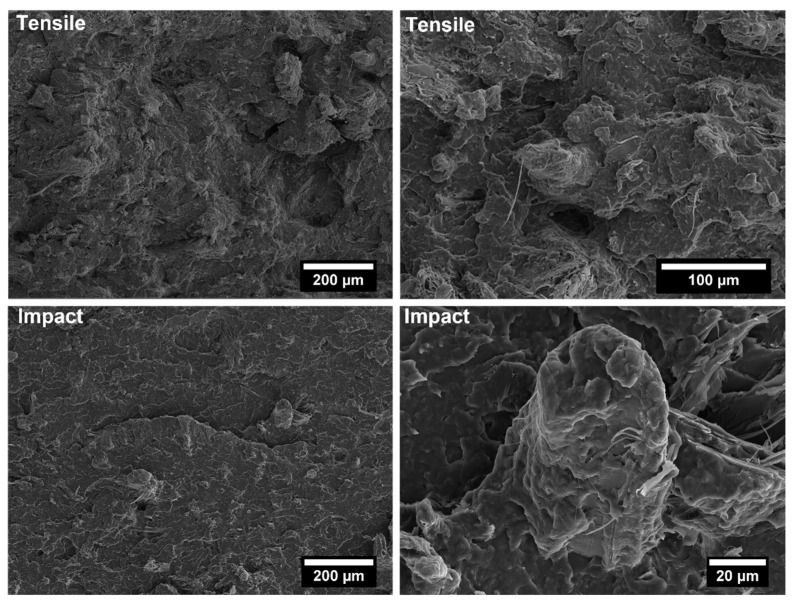
SEM micrographs of PP-based biocomposites with 30 wt.% of willow fillers with coupling agent after tensile and Charpy impact tests.

**Figure 17 polymers-14-04348-f017:**
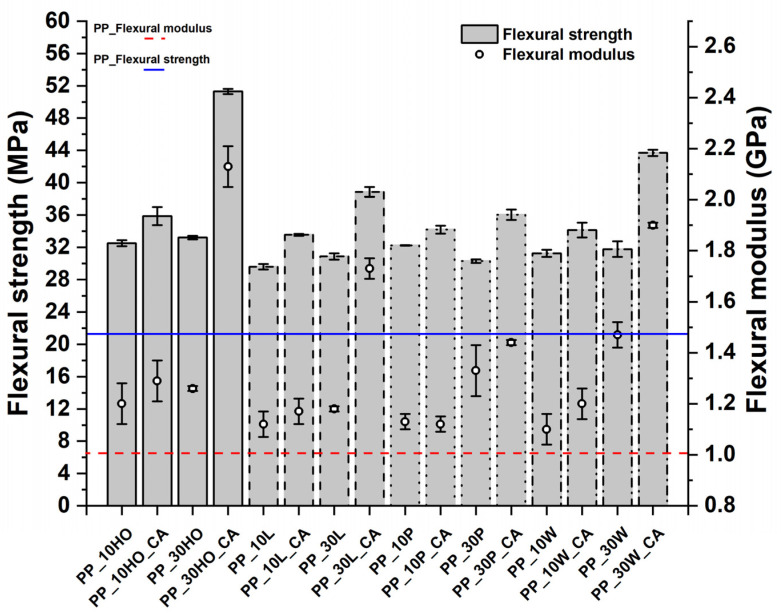
Flexural strength and modulus values of PP-based composites.

**Figure 18 polymers-14-04348-f018:**
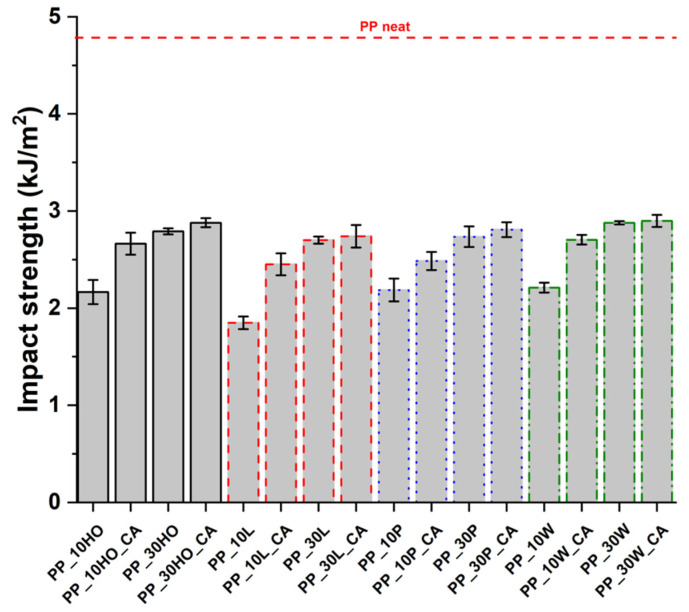
The results of Charpy impact tests for PP-based composites.

**Table 1 polymers-14-04348-t001:** Fractions of cellulose, lignin, and hemicellulose in biomasses, as extracted.

Biomass	Cellulose ^a^ (%)	Lignin ^a^ (%)	Hemicellulose ^a^ (%)
Palm Leaf ^b^ (P)	26	30	12
Licorice Root ^b^ (L)	44	12	15
Willow (W)	46	12	12
Holm Oak (HO)	40	13	14

^a^ all % are expressed as enriched fractions. ^b^ These values were reported in our previous studies [15].

**Table 2 polymers-14-04348-t002:** List of PP-based composites.

Formulation	PP (wt.%)	CA (wt.%)	Licorice Root (wt.%)	Palm Leaf (wt.%)	Willow (wt.%)	Holm Oak (wt.%)
PP	100	-	-	-	-	-
PP_10L	90	-	10	-	-	-
PP_10L_CA	85	5	10	-	-	-
PP_30L	70	-	30	-	-	-
PP_30L_CA	65	5	30	-	-	-
PP_10P	90	-	-	10	-	-
PP_10P_CA	85	5	-	10	-	-
PP_30P	70	-	-	30	-	-
PP_30P_CA	65	5	-	30	-	-
PP_10HO	90	-	-	-	-	10
PP_10HO_CA	85	5	-	-	-	10
PP_30HO	70	-	-	-	-	30
PP_30HO_CA	65	5	-	-	-	30
PP_10W	90	-	-	-	10	-
PP_10W_CA	85	5	-	-	10	-
PP_30W	70	-	-	-	30	-
PP_30W_CA	65	5	-	-	30	-

**Table 3 polymers-14-04348-t003:** Thermal degradation parameters for the different natural fillers.

Lignocellulosic Filler	T_d5_ (°C) ^a^	T_d10_ (°C) ^b^	T_max_ (°C) ^c^
Palm leaf	254.4 ± 0.9	275.5 ± 0.8	335.4 ± 0.7
Licorice root	269.1 ± 1.1	295.9 ± 0.9	355.3 ± 1.1
Holm oak	262.1 ± 0.9	285.3 ± 1.2	358.5 ± 0.8
Willow	269.7 ± 0.7	289.8 ± 0.9	354.3 ± 1.2

^a^ Temperature at 5% weight loss, ^b^ temperature at 10% weight loss, ^c^ temperature at maximum degradation rate.

**Table 4 polymers-14-04348-t004:** Thermal parameters obtained from Figure 5.

Sample	T_d5_ (°C)	T_d10_ (°C)	T_max_ (°C)
PP	423.7 ± 0.5	436.1 ± 0.6	464.8 ± 0.6
PP_10W	354.1 ± 0.9	407.5 ± 1.1	468.2 ± 0.9
PP_10W_CA	349.2 ± 1.2	399.7 ± 0.8	467.8 ± 1.3
PP_30W	324.4 ± 0.9	359.4 ± 0.6	468.3 ± 1.2
PP_30W_CA	306.6 ± 1.3	342.5 ± 0.9	469.3 ± 1.1
PP_10HO	389.3 ± 1.0	427.2 ± 0.8	466.8 ± 0.9
PP_10HO_CA	371.6 ± 1.2	422.5 ± 1.1	466.8 ± 0.7
PP_30HO	361.7 ± 1.2	419.0 ± 0.9	466.8 ± 1.3
PP_30HO_CA	317.2 ± 1.1	359.5 ± 1.3	466.5 ± 0.9
PP_10P	358.8 ± 0.9	424.9 ± 1.2	466.3 ± 1.4
PP_10P_CA	349.6 ± 0.7	421.5 ± 1.1	466.4 ± 1.2
PP_30P	321.8 ± 0.9	385.4 ± 0.9	468.1 ± 0.9
PP_30P_CA	299.1 ± 0.8	337.9 ± 0.8	469.4 ± 0.9
PP_10L	349.9 ± 1.2	402.1 ± 1.3	468.8 ± 1.3
PP_10L_CA	354.7 ± 1.3	408.7 ± 1.2	468.2 ± 0.9
PP_30L	312.5 ± 1.1	344.1 ± 1.4	472.3 ± 0.8
PP_30L_CA	308.9 ± 1.2	342.8 ± 1.3	473.2 ± 0.9

**Table 5 polymers-14-04348-t005:** Thermal properties of composites obtained from DSC analysis.

Sample	T_c_ (°C)	T_m_ (°C)	*X_c_* (%)
PP	116.6 ± 0.3	152.0 ± 0.3	36.3 ± 0.1
PP_10W	114.4 ± 0.6	150.3 ± 0.3	37.5 ± 0.3
PP_10W_CA	108.6 ± 0.4	149.5 ± 0.4	38.4 ± 0.2
PP_30W	108.1 ± 0.2	149.0 ± 0.4	42.5 ± 0.1
PP_30W_CA	107.5 ± 0.4	149.0 ± 0.6	40.0 ± 0.3
PP_10HO	115.8 ± 0.1	151.5 ± 0.3	40.3 ± 0.2
PP_10HO_CA	110.0 ± 0.4	149.1 ± 0.5	40.7 ± 0.2
PP_30HO	111.9 ± 0.8	150.4 ± 0.7	50.8 ± 0.5
PP_30HO_CA	108.1 ± 0.2	149.2 ± 0.3	44.3 ± 0.7
PP_10P	109.4 ± 0.6	149.2 ± 0.3	36.6 ± 0.5
PP_10P_CA	108.6 ± 0.1	149.0 ± 0.1	38.5 ± 0.2
PP_30P	107.1 ± 0.2	148.0 ± 0.4	44.6 ± 0.6
PP_30P_CA	109.7 ± 0.6	148.6 ± 0.6	39.6 ± 0.2
PP_10L	106.2 ± 0.3	148.2 ± 0.6	35.3 ± 0.3
PP_10L_CA	107.8 ± 0.3	149.4 ± 0.6	36.4 ± 0.3
PP_30L	107.0 ± 0.6	147.2 ± 0.5	31.5 ± 0.6
PP_30L_CA	108.0 ± 0.3	148.9 ± 0.2	33.4 ± 0.1

**Table 6 polymers-14-04348-t006:** Glass transition temperature (T_g_) and storage modulus values (E’) of PP-based composites at different temperatures.

Sample	T_g_ (°C)	E’@−50 °C (MPa)	E’@0 °C (MPa)	E’@25 °C (MPa)	E’@50 °C (MPa)	E’@80 °C (MPa)
PP	3.2 ± 0.1	4380.2 ± 16.4	2330.5 ± 17.2	1262.8 ± 14.5	712.2 ± 14.1	298.4 ± 11.8
PP_10W	−2.2 ± 0.1	4917.3 ± 13.4	2085.4 ± 9.7	1230.3 ± 12.4	739.8 ± 12.1	350.7 ± 12.6
PP_10W_CA	−2.3 ± 0.1	4840.6 ± 20.1	2095.3 ± 11.1	1243.4 ± 10.8	752.3 ± 11.4	362.4 ± 11.9
PP_30W	−0.4 ± 0.1	5399.4 ± 11.8	2609.1 ± 12.8	1614.8 ± 13.1	1023.8 ± 12.1	506.7 ± 12.1
PP_30W_CA	−2.3 ± 0.1	5529.4 ± 12.1	2653.8 ± 11.4	1674.4 ± 12.1	1077.4 ± 11.4	554.8 ± 12.2
PP_10HO	2.7 ± 0.1	4785.7 ± 12.8	2420.7 ± 10.9	1342.4 ± 10.2	763.2 ± 11.1	361.4 ± 9.9
PP_10HO_CA	0.3 ± 0.1	4836.8 ± 13.1	2486.4 ± 12.8	1401.7 ± 13.1	800.9 ± 11.9	370.7 ± 11.4
PP_30HO	0.3 ± 0.1	4810.3 ± 14.7	2436.4 ± 11.4	1347.8 ± 13.9	785.6 ± 11.7	368.4 ± 12.2
PP_30HO_CA	0.4 ± 0.1	5136.4 ± 12.6	2656.3 ± 11.8	1596.2 ± 12.9	999.4 ± 10.9	510.4 ± 12.4
PP_10P	0.4 ± 0.1	4776.1 ± 11.1	2241.7 ± 10.8	1249.4 ± 13.4	736.8 ± 12.1	334.4 ± 11.9
PP_10P_CA	1.4 ± 0.1	4848.7 ± 10.8	2367.4 ± 11.7	1367.8 ± 12.1	814.7 ± 13.1	374.4 ± 12.9
PP_30P	−0.6 ± 0.1	4953.1 ± 11.7	2371.4 ± 12.8	1378.1 ± 10.9	828.4 ± 12.1	396.5 ± 11.9
PP_30P_CA	−0.5 ± 0.1	5394.1 ± 11.9	2688.4 ± 12.6	1630.7 ± 11.9	1018.7 ± 13.4	512.4 ± 10.9
PP_10L	−0.5 ± 0.1	4627.4 ± 10.8	2051.8 ± 9.9	1168.4 ± 11.7	698.4 ± 10.8	324.4 ± 11.7
PP_10L_CA	−0.5 ± 0.1	4890.4 ± 11.9	2150.3 ± 10.9	1231.4 ± 12.7	756.4 ± 11.1	346.8 ± 12.3
PP_30L	−2.6 ± 0.1	6152.3 ± 11.2	3023.8 ± 11.4	1902.8 ± 10.9	1203.7 ± 12.8	597.9 ± 12.6
PP_30L_CA	−1.4 ± 0.1	6680.7 ± 12.7	3444.9 ± 12.6	2302.7 ± 11.9	1504.2 ± 13.1	804.7 ± 11.9

## Data Availability

The data presented in this study are available on request from the corresponding author.

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
