# Peer review of "Chemical, Thermal and Mechanical Characterization of Licorice Root, Willow, Holm Oak, and Palm Leaf Waste Incorporated into Maleated Polypropylene (MAPP)"

_polymers, 2022, doi:10.3390/polym14204348_

Round 1

Reviewer 1 Report

The manuscript addresses the combination of different less conventional natural fiber powders in different ratios and in combination with a thermoplastic by extrusion followed by injection molding. In addition, the typical coupling agent based on maleic anhydride it was also used to improve the mechanical properties. The work uses adequate characterization techniques and it is presented in a comprehensive manner, however there are some comments that should be addressed before to be considered for publication, namely:

Pages 6 and 7: For the morphology analysis of Figure 3, 4, 5 and 6, those figures can be grouped in only one figure being each waste fibre represented with two different magnifications. Moreover, it is recommended to provide the used magnification in the caption.

Page 8, Figure 7: Based on the results obtained for TGA analysis and considering the processing conditions (i.e. barrel temperature of 215 °C and a die temperature of 210 °C, followed by injection molded process at 200ºC), it is not clear that the decrease of mass in TGA of the natural fibres it is only due to the moisture content. Moreover, and considering the used two melt-based processes, why the author selected to monitor the thermal stability under nitrogen (inert atmosphere) and not under air? In the manuscript, it is not clear if the fiber was pre-dried or not before the TGA analysis.

Page 14, line 392: The author states that “All composite formulations displayed a much more macroscopic brittle behavior”, however and based on the stress–strain curves displayed in Figure 11, it seems that one percentage of the produced composites presents a ductile fracture behavior. This paragraph should be more clear.

Page 15, table 7: Regarding the “mechanical properties of polymer matrix composites reinforced with natural filler” the reviewer recommends to consider a manuscript using a different natural fiber that was combined with polypropylene matrix and also includes the use of maleic anhydride as adhesion promoter and using similar processing ( https://doi.org/10.1016/j.compositesb.2014.05.019 ). The reference might be also useful for improve the discussion section.

Page 21, line 493: In the conclusion section, the author states that “A reduction in β-relaxation peak area with increasing filler”. This type of events can be observed and confirmed by other characterization techniques, thus the reviewer recommends to revise the sentence by indicating the used technique DMA.

Author Response

The manuscript addresses the combination of different less conventional natural fiber powders in different ratios and in combination with a thermoplastic by extrusion followed by injection molding. In addition, the typical coupling agent based on maleic anhydride it was also used to improve the mechanical properties. The work uses adequate characterization techniques and it is presented in a comprehensive manner, however there are some comments that should be addressed before to be considered for publication, namely:

Pages 6 and 7: For the morphology analysis of Figure 3, 4, 5 and 6, those figures can be grouped in only one figure being each waste fibre represented with two different magnifications. Moreover, it is recommended to provide the used magnification in the caption.

Ans.: The authors thank the reviewer for the suggestion. Figures from 3 to 6 have been grouped in one figure, which includes two different magnifications for each filler. The magnification, despite not being very informative without the corresponding working distance, has been added in each micrograph.

Page 8, Figure 7: Based on the results obtained for TGA analysis and considering the processing conditions (i.e. barrel temperature of 215 °C and a die temperature of 210 °C, followed by injection molded process at 200ºC), it is not clear that the decrease of mass in TGA of the natural fibres it is only due to the moisture content. Moreover, and considering the used two melt-based processes, why the author selected to monitor the thermal stability under nitrogen (inert atmosphere) and not under air? In the manuscript, it is not clear if the fiber was pre-dried or not before the TGA analysis.

Ans.: The authors thank the reviewer for the remark. The lignocellulosic fillers were not pre-dried before performing the TGA analysis, and this explains the occurrence of the first mass loss at T lower than 150 °C, where free water evaporates at lower temperature while the bound water from chemical bonds with the hydroxyl groups present in hemicellulose and cellulose evaporates at higher temperatures (https://doi.org/10.1016/S0040-6031(99)00471-2). After this first mass loss step, of course the degradation followed the common trend including hemicellulose and cellulose degradation with two distinct peaks. The thermal stability of fillers was investigated in nitrogen for two main reasons: (i) to avoid potential oxidation reactions of cellulose and (ii) to compare these results with those of other fillers, being very common in literature the use of inert atmosphere (https://doi.org/10.1016/j.compositesb.2012.07.027; 10.1016/j.polymdegradstab.2010.02.009), thus avoiding potential shifts of DTG peaks at lower temperatures in the  oxidative atmosphere compared to the inert one (https://doi.org/10.1016/j.msea.2012.05.109; https://doi.org/10.3390/polym13162710). The text has been modified accordingly.

Page 14, line 392: The author states that “All composite formulations displayed a much more macroscopic brittle behavior”, however and based on the stress–strain curves displayed in Figure 11, it seems that one percentage of the produced composites presents a ductile fracture behavior. This paragraph should be more clear.

Ans.: The reviewer is right: the sentence was misleading. Of course, composites at 10 wt.% still showed a ductile behavior, but this tendency was reduced by increasing filler content above 10 wt.% and by adding the coupling agent. The text has been modified accordingly.  

Page 15, table 7: Regarding the “mechanical properties of polymer matrix composites reinforced with natural filler” the reviewer recommends to consider a manuscript using a different natural fiber that was combined with polypropylene matrix and also includes the use of maleic anhydride as adhesion promoter and using similar processing ( https://doi.org/10.1016/j.compositesb.2014.05.019 ). The reference might be also useful for improve the discussion section.

Ans.: The authors thank the reviewer for the useful suggestion. Despite similarities in the matrix, coupling agent and processing strategies, the authors decided not to include results of this study as cork is characterized by a different chemical composition with respect to the investigated lignocellulosic fillers, mostly based on suberin and lignin, with limited amount of cellulose and hemicelluloses, thus stressing the much less important role of cellulose in cork (10.15376/biores.10.3.Pereira).

Page 21, line 493: In the conclusion section, the author states that “A reduction in β-relaxation peak area with increasing filler”. This type of events can be observed and confirmed by other characterization techniques, thus the reviewer recommends to revise the sentence by indicating the used technique DMA.

Ans.: The authors thank the reviewer for the remark. The conclusions section has been modified including the fact that the “reduction in β-relaxation peak area with increasing filler content” was obtained by DMA technique.

Reviewer 2 Report

Fig.20: Consider shifting the legend scale either for "Flex. strength" or "Flexural modulus" as some markers in the graph are overlapping,  and make it difficult to interpret the results.

Author Response

Reviewer 2

Fig.20: Consider shifting the legend scale either for "Flex. strength" or "Flexural modulus" as some markers in the graph are overlapping,  and make it difficult to interpret the results.

Ans.: The figure has been modified accordingly.

Reviewer 3 Report

The paper can be considered as a benchmarking review of lignocellulosic fillers. To be considered as a contribution to the scientific journal, author must improve the description of new knowledge. Additionally, the observations and comments in the attached PDF must be considered.  

Author Response

Reviewer 3

The paper can be considered as a benchmarking review of lignocellulosic fillers. To be considered as a contribution to the scientific journal, author must improve the description of new knowledge. Additionally, the observations and comments in the attached PDF must be considered.

Ans.: “die temperature” represents the temperature at the end of the extruder, therefore it is not clear what needs to be changed. As regards former figures from 3 to 6 (noe reduced to one figure as per reviewer 1 comment), all the elements included in the micrographs represent the fillers, as these SEM pictures were taken only for the fillers themselves. “Derivative” means the first derivative of the weight with respect to time, as it is generally used to highlight points of greatest rate of change on the weight loss curve.

Of course, SEM micrographs cannot represent the existence of a covalent bond, but instead can show a stronger filler/matrix interfacial adhesion because of the covalent bonds due to the coupling agent, as reported in the references. The sentence was slightly modified to include the reviewer’s comment.